# The Analysis of the Correlations between BMI and Body Composition among Children with and without Intellectual Disability

**DOI:** 10.3390/children9050582

**Published:** 2022-04-20

**Authors:** Bogdan Constantin Ungurean, Adrian Cojocariu, Beatrice Aurelia Abalașei, Lucian Popescu, Alexandru Rares Puni, Marius Stoica, Carmen Pârvu

**Affiliations:** 1Faculty of Physical Education and Sports, “Alexandru Ioan Cuza” University of Iași, 507184 Iași, Romania; beatrice.abalasei@uaic.ro (B.A.A.); lucian_popescu2009@yahoo.com (L.P.); punirares@yahoo.com (A.R.P.); 2Doctoral School in Sports and Physical Education Science, Faculty of Physical Education and Sports, “Alexandru Ioan Cuza” University of Iasi, 507184 Iași, Romania; cadriano@uaic.ro; 3Faculty of Physical Education and Sports, National University of Physical Education and Sport, 060057 București, Romania; mariusstoica08@yahoo.com; 4Faculty of Physical Education and Sport, “Dunarea de Jos” University of Galati, Romania, 337347 Galati, Romania; carmen_preda06@yahoo.com

**Keywords:** body mass index, body composition, intellectual disability

## Abstract

Background: Compared to the great volume of studies focusing on children and adolescents without intellectual disability, research regarding body mass index among young populations (13–17 years old) with intellectual disability is scarce, mostly when we refer to the comparisons between various degrees of intellectual disability and gender. Methods: The purpose of this study was to assess a series of morphofunctional parameters among children with and without intellectual disability to characterise the morphofunctional normality and its perturbations. Within the study, we included 101 subjects from several educational institutions, distributed on five groups, by their gender and degree of intellectual disability. Results: The average values of body mass index exceed the values recommended by the WHO among all the five groups (boys and girls with and without intellectual disabilities) prone to obesity. Upon analysing the values of BMI by gender and type of intellectual disability, we note that the prevalence of obesity among boys is 28.07% (BMI > 24), while 19.29% are overweight (BMI ranging between 21.5 and 24). Conclusions: The prevalence of excess weight and obesity among persons with intellectual disabilities was similar among the male and female subjects. It shows an increasing trend by age.

## 1. Introduction

People with disabilities are 2.5 times more likely of being inactive than people without disabilities, while youths with disabilities are 4.5 times more inactive than youths without disabilities [1,2]. Around 47% of the adults with disabilities are obese, accounting for 10% more than adults without disabilities.

Generally, the activity level of people with disabilities is inadequate to prevent illness and other chronic diseases caused by inactivity [3,4,5,6]. International statistics show that the number of people with disabilities has been recording an ascending trend worldwide. The World Health Organisation (WHO), in collaboration with the World Bank (WB), launched on 9 June 2011 the first global report on disability [7].

The studies and research conducted to draft up the report have shown that over a billion people worldwide have some sort of disability; this accounts for about 15% of the world population. The WHO had published its most recent statistics concerning disability in the 1970s, when it estimated around 10% of the world population as being affected. The WHO report made in collaboration with the World Bank suggests that the number of people with disabilities may increase in the near future, mostly due to population ageing, as the elderly are more prone to disability. Another cause is the increase in the frequency of chronic diseases such as diabetes, cardiovascular conditions, cancer, and mental health disorders.

A recent study [8] shows that the available evidence on the country-based variation in exposure to established determining factors of intellectual disability (i.e., poverty, malnutrition) suggests a higher incidence of intellectual disability in low-income countries.

In the European Union (EU), the number of people with disabilities is 80 million, accounting for around 15% of the EU population; 1 in 4 Europeans has a family member suffering from a disability [9]. Among children without intellectual disabilities, many health benefits have been associated with physical activity, including psychosocial benefits [10], balance, muscle power, quality of life [11], neurocognitive function, inhibiting control [12], and better motor activity [13,14]. However, the physical activity level among children and adolescents with intellectual disability is low [2,15].

Excess weight and obesity among children represent one of the most serious public health issues of our time. In the 1980s, excess weight rates among children increased suddenly. Worldwide, in the four decades between 1978 and 2017, the number of children aged between 5 and 19 with a body mass index (BMI) categorised as obese increased 10 times, from 11 million to 124 million [16]. The prevalence of childhood obesity has doubled in the last 30 years among pre-schoolers and adolescents, and it has tripled for the age group of 6–11 years old. To define obesity among people aged 2–19, we use the following terms: overweight if BMI (body mass index) ranges between the 85th and the 95th percentiles for their age and obesity if BMI is above the 95th percentile for their age [17]. Child obesity is a predictive factor for morbidity and mortality at adult age: Up to 80% of obese children will be obese adults, highly prone to cancer, high blood pressure, strokes, liver and bile disease, and osteoarthritis [18]. Recently, studies carried out in the United States have shown moderate or intense physical activity to correlate with a lower body mass index and a lower overweight incidence, though we believe that the direct association between the time spent watching TV and overweight is even more relevant.

The purpose of our research is to identify the parameters of body composition that influence the values of the body mass index in children with and without intellectual disability.

**Hypothesis** **1.**
*Body mass index is influenced by the values of body fat and basal metabolic rate in children with and without intellectual disability.*


**Hypothesis** **2.**
*There are effects of interaction between the variables of intellectual disability and gender on the components of body mass composition.*


## 2. Materials and Methods 

The research objectives are to assess a series of morphofunctional and motor parameters among children with and without intellectual disability and analyse the data obtained from the statistical-mathematical indicators in relation to the scientific literature.

Ethics: We conducted this study pursuant to the Declaration of Helsinki.

Morphofunctional parameters:

Height—For a correct measurement of the subjects’ height, they took their shoes off and they stood up, touching a vertical wall with their back, head, and heels, head facing forward. Using a telemeter, we measured the distance from the floor to the perpendicular wall projection of the vertex point level (the highest cranial point), determined using an item featuring an 90° angle (e.g., a protractor) with one side placed on the vertex and the other on the wall. We recorded the value in centimetres and 0.5 cm subdivisions. Within the measurements made on this group of subjects, we used a Bosch GLM 80 laser telemeter for an accurate value.

To determine body composition, we used a professional TANITA MC 580 device and a dedicated analysis software GMON V3.4 [19].

The bioelectrical impedance analysis (BIA) is a technique used to measure body composition. Bioelectrical impedance analysis technology involves a low-intensity electrical current (around 500 µA) travelling from the electrodes underneath the soles to the electrodes held in both hands. Professional models provide a segment-based analysis: The six electrodes show additional information for each leg, arm, and area (abdominal). The electrical signal travels rapidly through the water present in the lean, hydrated tissue, but it has a harder time travelling through the adipose tissue. This resistance, known as impedance, is measured and included in the scientifically validated Tanita equations to calculate body composition measurements. The TANITA multifrequency monitors can measure the bioelectrical impedance analysis on three to six different frequencies. Additional frequencies provide an exceptional accuracy level compared to monitors featuring one or two frequencies. Lower frequencies measure impedance outside the cell membrane. Higher frequencies can penetrate the cell membrane, thus measuring lower and higher impedance. Therefore, we can estimate extracellular and intracellular water, as well as total body water. All this information is essential to provide data on the health state of a person and indicate potential health risks [20].

TANITA PRO SOFTWARE—The Tanita PRO software package was developed in partnership with the most relevant medical software developer [21]. This software can store and analyse data from the Tanita MC 580 monitor. In conformity with the EU regulations, the software has medical certification, and its meets all the Regulations into force (Council Directive 93/42/EEC of 14 June 1993 concerning medical devices).

The use of TANITA MC580 and TANITA PRO SOFTWARE generates 11 measurements: body mass (weight)—Kg; BMI (kg/h^2^); body fat %; muscle mass %; RMB (kcal); body fat—Kg; muscle mass—Kg; SMM—skeletal muscle mass; total water; bone mineral mass; segmental analysis by upper/lower limbs, left/right.

Basal metabolic rate (BMR)—Basal metabolic rate is an estimate, automatically generated by the TANITA PRO software, of the minimum number of calories that a person needs on a daily basis to maintain basic functions (breathing, circulation, and digestion) at rest.

We began the measurements in April 2021, and we continued them until November 2021, depending on the pandemic context (COVID-19) of the period, and these were conducted in the morning between 10 a.m. and 12 p.m. We carried out the activities in the gyms of the educational institutions and in the physical therapy practices of the “Sf. Andrei” School Centre Gura Humorului and the “Constantin Păunescu” School Centre Iaşi.

We introduced and analysed software the results obtained through the measurements in IBM SPSS 20.0 using the following statistical tests: ANOVA Multifactorial (the post-hoc analysis included, to decide on the possible significant differences between the parameters measured), Tukey HSD (honestly significant difference-HSD) used for uneven samples, as well as PEARSON’s correlation. Considering sample sizes (smaller than 50 subjects), to test data distribution normality, we used the Shapiro–Wilk test, which is more accurate than the Kolmogorov–Smirnov test.

Within the study, we included 101 subjects from several educational institutions distributed in five groups, as illustrated in Table 1.

## 3. Results

After applying the distribution normality test (Shapiro–Wilk) by groups, we note that in the group of boys without ID, the only significant differences from the normal distribution (*p* = 0.048) concerns body fat in kilograms (Table 2). For the group of girls without ID (Table 2), we found significant differences from the normal curb concerning several dependent variables, as follows: body mass (*p* = 0.012) and body fat in kg (*p* = 0.006). In the group of boys with MID (Table 2), significant differences from the normal distribution were found in BMI (*p* = 0.005), muscle mass in percentages (*p* < 0.001), and body fat in kilograms (*p* = 0.003). In the group of girls with MID, we note two variables without a normal distribution, as follows: height (*p* < 0.001) and percentage muscle mass, where *p* = 0.007 (Table 2). In what concerns the group of boys with SID, the only significant differences relate to age *p* < 0.05 (Table 2). Following the application of the Levene test, we identified that in all dependent variables the variances are assumed to be equal (comparing the groups of boys-group 1, group 3, group 5), including in situations where the distribution is not normal: body fat (kg) in group 1 and BMI, muscle Mass %, and body fat Kg in group 3. Regarding these variables, the post-hoc analysis of the differences between the three groups cannot be interpreted categorically, although no significant differences were identified.

Following the analysis of body mass index values (Table 3), we note that the minimum value among boys is featured in the SID group (16.0 kg/h^2^), while the maximum value concerns the MID group (34.7 kg/h^2^). In the groups of girls, both the minimum value (15.9 kg/h^2^) and the maximum value (34.2 kg/h^2^) concern the group without ID. Upon analysing the average values for the groups of girls, it may be noted that a BMI of 21.58 kg/h^2^ in the group without ID and of 21.32 kg/h^2^ in the MID group ranges in the limits of the WHO values. Regarding BMI in the groups of boys, its mean value is 22.95 kg/h^2^ in the group without ID, 22.53 kg/h^2^ in the MID group, and 22.33 kg/h^2^ in the SID group.

It is worth noting that in both the group of boys with MID and the group of girls with MID, there is a weak negative correlation (r = −0.288 and r = −0.278, respectively) between body mass index and the percentage of muscle mass. In other words, the higher the BMI, the lower the percentage muscle mass, but not as much as in the groups of boys without ID and girls without ID, where r = −0.817 (*p* < 0.001) and r = −0.799 (*p* < 0.001), respectively.

Basal metabolic rate, generated automatically by the Tanita MC580 platform through the Tanita Pro software, has values ranging between a maximum of 2726 Kcal among boys in the group without ID and a minimum of 1368 Kcal in the SID group. For the groups of girls, the highest value (1726 Kcal) and the lowest value (1170 Kcal) concerning the group without ID (Table 3).

Upon applying the ANOVA test, through the multiple comparison method Tukey’s HSD (used for uneven samples, for the three groups of boys), we have found a strong significant threshold, *p* < 0.05, for the comparison between the group of boys without ID and the group of boys with SID, where *p* = 0.004 (Table 4).

Upon analysing the values of Pearson’s coefficient correlation for the group of boys without ID, we point out (Annex 1) that for r = 0.888 (*p* < 0.001), there is a very strong positive correlation between BMI and percentage body fat (Figure 1) and a very strong negative correlation for r = −0.817 (*p* < 0.001) between BMI and muscle mass percentage (Figure 2).

For the group of boys with MID, we find the same very strong positive correlation between BMI and body fat in percentages r = 0.809 (*p* < 0.001) (Annex 2, Figure 3), but concerning the correlation between BMI and the percentage muscle mass, there is a weak negative correlation, for r= −0.288 (Annex 2, Figure 4).

For the group of boys with SID, we found an extremely strong positive correlation (Graph 1 and Annex 7 Appendix A) between body mass index and percentage body fat r = 0.983 (*p* < 0.001) (Annex 3) and an extremely strong negative correlation between body mass index and percentage muscle mass (Graph 2 and Annex 7 Appendix A) for r = −0.982 (*p* = 0.001) (Annex 3).

In the group of girls without ID, there are the same types of correlations between body mass index and percentage body fat (strong positive correlation for r = 0.797 (*p* < 0.001), Figure 5) and between body mass index and percentage muscle mass (strong negative correlation for r = −0.799 (*p* < 0.001), Figure 6 (Annex 4).

In the group of girls with MID, the correlation regime follows the same direction, with a strong positive correlation (Graph 3, Annex 7 Appendix A) between body mass index and percentage body fat for r = 0.820 (*p* = 0.001), and a weak negative correlation (Graph 4, Annex 7 Appendix A) between body mass index and percentage muscle mass, for r = −0.278. (Annex 5)

For the groups of boys, we have found a very high correlation between BMR and muscle mass expressed in kilograms (Figure 7), for r = 979, *p* < 0.001 (in the group without ID); r = 940, *p* < 0.001 (Figure 8—the MID group), and a high correlation for the SID group for r = 0.776 (Annex 1, Annex 2, Annex 3).

Concerning the groups of girls, we note a very high correlation between BMR and muscle mass expressed in kilograms in the group without ID (Graph 5, Annex 7 Appendix A) for r = 0.963, *p* < 0.001 and a high correlation for the MID group (Graph 6, Annex 7 Appendix A) for r = 0.771 *p* < 0.05 (Annex 4, Annex 5).

Upon analysing the results concerning skeletal muscle mass (SMM), it is worth noting that, in the three groups of boys, we found very high correlations between SMM and muscle mass in kilograms, as follows: r = 0.979 (*p* < 0.001) for the group of boys without ID (Annex 1); r = 1.00 (*p* < 0.001), which represents a perfect correlation between the two variables for the group of boys with MID (Figure 9, Annex 2); and r = 0.995 (*p* < 0.001) for the group of boys with SID (Annex 3).

In the groups of girls, there is a perfect correlation (Figure 10) between SMM and muscle mass in kilograms for r = 1.00 (*p* < 0.001) (Annex 4) in the group of girls without ID, while in the group of girls with MID, we found a high correlation for r = 0.799 (Annex 5).

Upon analysing the correlations between the dependent variables considered in the study for the entire sample, we found a very high positive correlation (Figure 11) between body mass index and body fat expressed in kilograms for r = 0.824 (*p* < 0.001), as well as a negative correlation (Figure 12) between body mass index and percentage muscle mass r= −0.409 (*p* < 0.001) (Annex 6).

## 4. Discussion

The average values of body mass index exceed the values recommended by the WHO among all the five groups prone to obesity. Upon analysing the values of BMI by gender, we note that the prevalence of obesity among boys is 28.07% (BMI > 24), while 19.29% are overweight (BMI ranging between 21.5 and 24). In the groups of girls, we note that 15.78% have a BMI > 24 (obesity), while 39.47% are overweight, with a BMI ranging between 21.1 and 23.9. Unfortunately, it is impossible to find a connection between the findings we have obtained and Romanian scientific studies. The extension of the number of subjects in the groups could be the subject of future research, and perhaps this will allow us to obtain normal distributions in the case of Body fat (kg) in group 1 and BMI, muscle mass%, and body fat Kg in group 3 (boys with moderate intellectual disability). However, recent international studies [22] confirm the overweight/obesity propensity of people with ID and without ID, regardless of the gender [23,24,25]. In another recent study that used the BIA technology and a Tanita device, just like us [26], the authors identified high values of the BMI, exceeding the values recommended by the WHO among people with intellectual disabilities.

The prevalence of childhood obesity has doubled in the last 30 years for preschoolers and adolescents and has tripled for the 14–18 age group [27]. The following terms are used to define obesity in the 4–19 age group: overweight if the BMI is between the 85th and 95th percentiles for age and obesity if the BMI is above the 95th percentile for age [28]. Childhood obesity is a predictive factor for adult morbidity and mortality: Up to 80% of obese children will be obese adults, who, in addition to psychosocial disadvantages, have an increased susceptibility to cancer, high blood pressure, stroke, liver and bile duct disease, osteoarthritis, and diabetes [18,29]. Studies in the literature suggest that the physical activity of obese children, but especially of children with ID, is much lower compared to other children, but a strict relationship between physical activity and adiposity in the general population is more difficult to establish. Recently, studies have shown that intense physical activity, such as combat sports, [30] correlates with a lower body mass index and a low frequency of overweight. Exercise and dietary interventions result in more weight loss than diet alone. Lifestyle through exercise (running, swimming) is more effective in reducing weight in the long run compared to the aerobic or calisthenics exercises (exercises with your own body weight, for example pull-ups or push-ups). Interventions related to the decrease in sedentary behaviors seem to be more effective in lowering the body mass index than interventions related to the intensification of physical activity, although a recent study [31] shows that the value of the body mass index could not be associated with components of physical activity. It is important to note that physical activity seems to maintain a proper regulation of body weight associated with normal growth and maturation. Moreover, depending on the degree, duration, and distribution of adipose tissue in children, this condition is likely to continue into adulthood [32]. In the last two years (with the onset of the COVID-19 pandemic), recent studies have shown that a decrease in physical activity correlates with an increase in body mass index [33].

Some limitations of this study should be noted. First, the representativeness of the sample and, therefore, the generalization of the results are limitations. The second limitation is that due to the recent pandemic situation, access has been limited to schools for children with intellectual disabilities. The third limitation is given by limited bibliographic sources regarding the influence of intellectual disability on some parameters of body composition. The fourth limitation is the relatively small number of subjects, especially in the group of children with severe intellectual disabilities.

It should be noted that overweight and obesity affect and continue to affect all subgroups of the population, regardless of gender, age, IQ, ethnicity, socioeconomic status and life experience.

## 5. Conclusions

This study attempted to clarify the relationship between the degree intellectual disability and some morphological parameters among children with and without intellectual disability. Among other observations, we have noted a significant association between intellectual disability and a series of body composition parameters.

A strong point of this study is that it is one of the few examining the relationship between body composition among children with and without intellectual disability of various degrees, by gender.

The prevalence of excess weight and obesity among persons with intellectual disabilities was similar among the male and female subjects. It shows an increasing trend by age.

Whereas our findings represent a reliable starting point, additional research is necessary to clarify the existence of a correlation between the physical activity volume among people with ID and the parameters analysed within the study. If we assess and solve this uncertainty in an effective manner, we can develop physical activity programs to improve the abilities and quality of life among people with intellectual disability.

The findings of our study may help to set up intervention strategies for the management of obesity, if the decision makers prioritise the treatment of persons with intellectual disabilities.

## Figures and Tables

**Figure 1 children-09-00582-f001:**
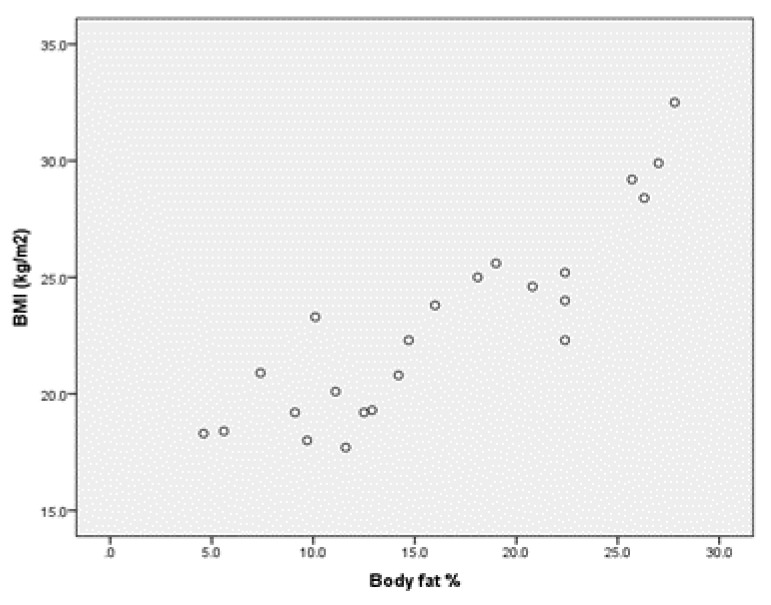
Correlation between BMI and body fat % for the group of boys without ID.

**Figure 2 children-09-00582-f002:**
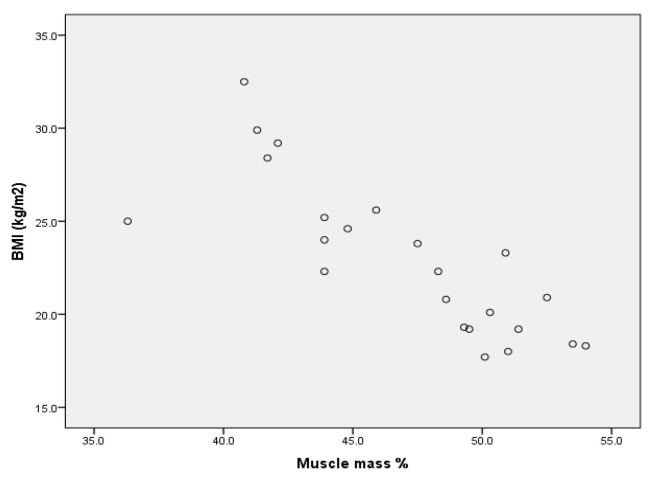
Correlation between BMI and muscle mass % for the group of boys without ID.

**Figure 3 children-09-00582-f003:**
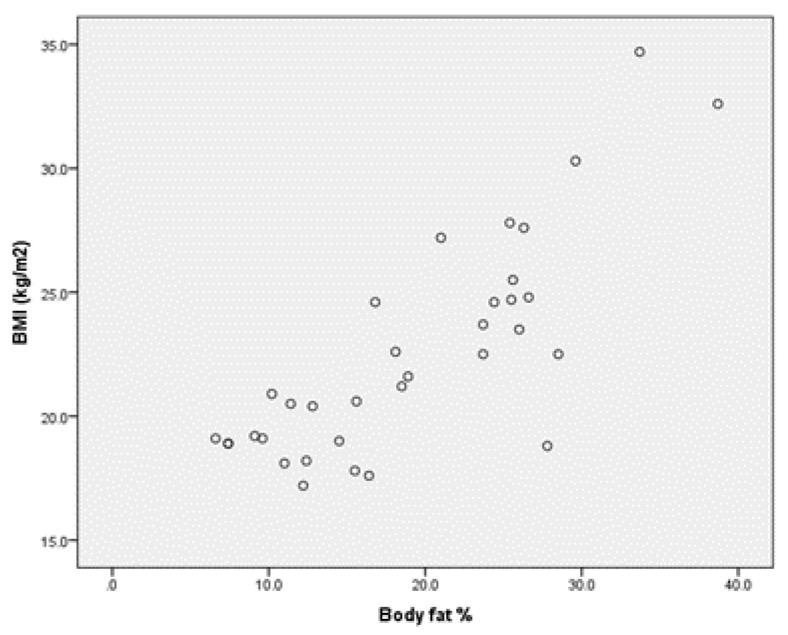
Correlation between BMI and body and fat % for the group of boys MID.

**Figure 4 children-09-00582-f004:**
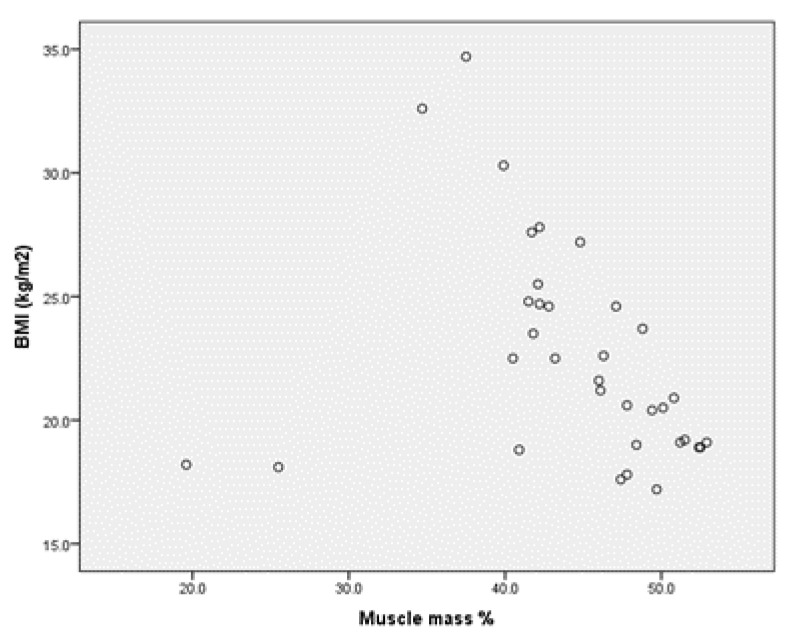
Correlation between BMI and muscle and mass % for the group of boys MID.

**Figure 5 children-09-00582-f005:**
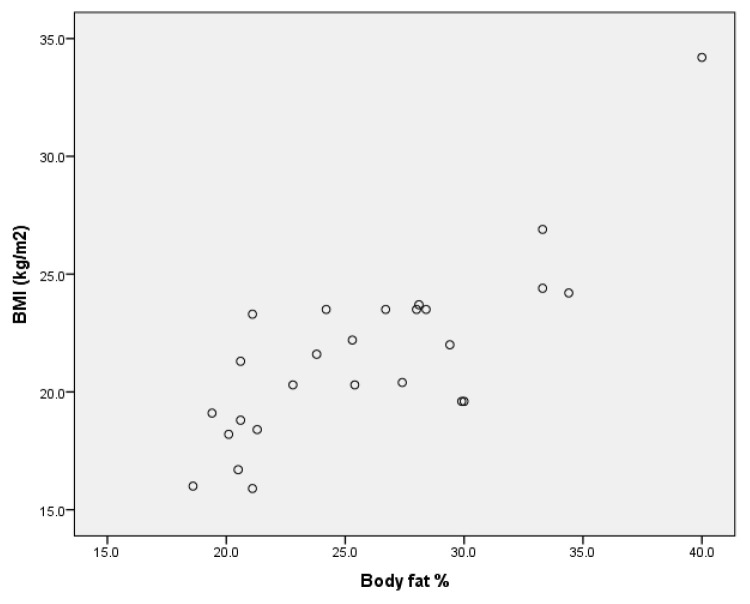
Correlation between BMI and body fat % for the group of girls without ID.

**Figure 6 children-09-00582-f006:**
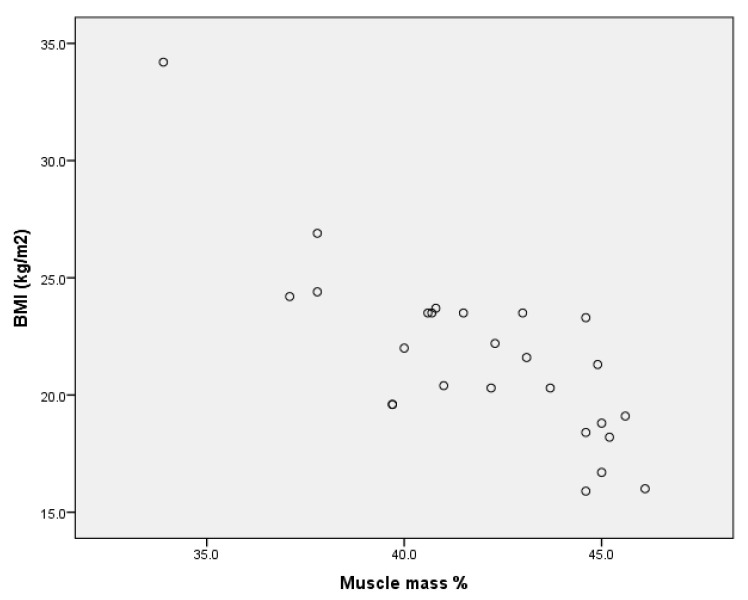
Correlation between BMI and muscle mass % for the group of girls without ID.

**Figure 7 children-09-00582-f007:**
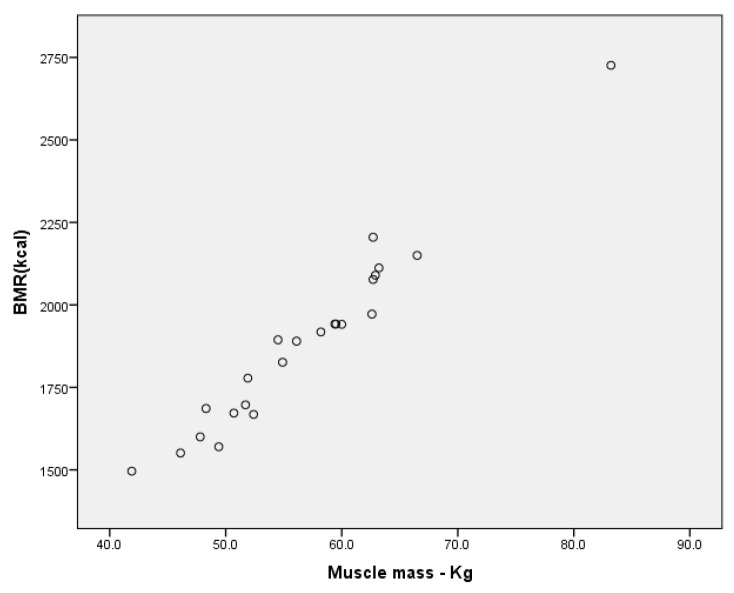
Correlation between BMR and muscle mass (kg) for the group of boys without ID.

**Figure 8 children-09-00582-f008:**
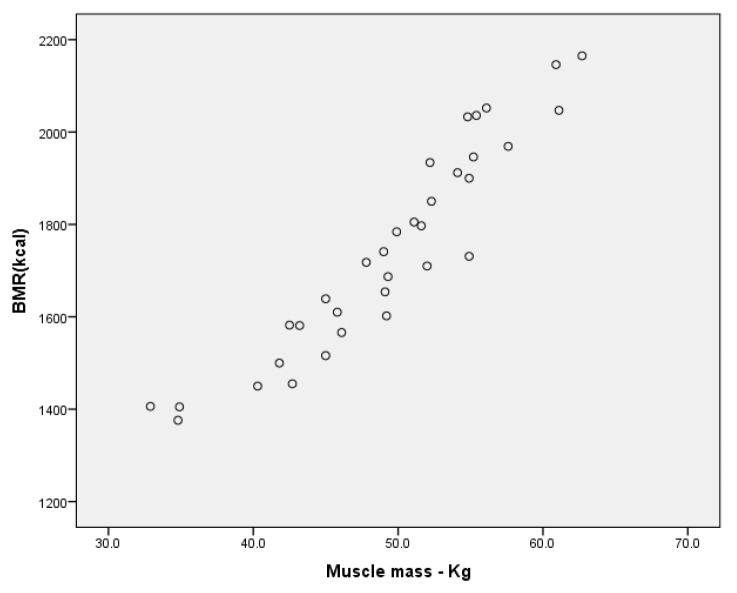
Correlation between BMR and muscle mass (kg) for the group of boys MID.

**Figure 9 children-09-00582-f009:**
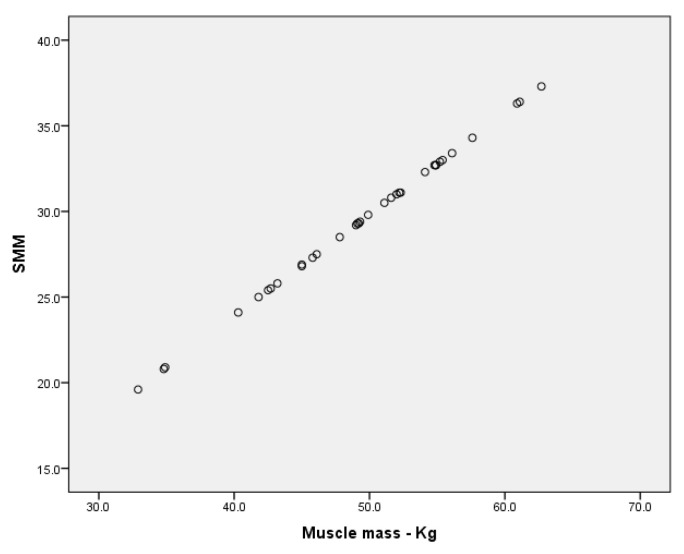
Correlation between SMM and muscle mass (kg) for the group of boys with MID.

**Figure 10 children-09-00582-f010:**
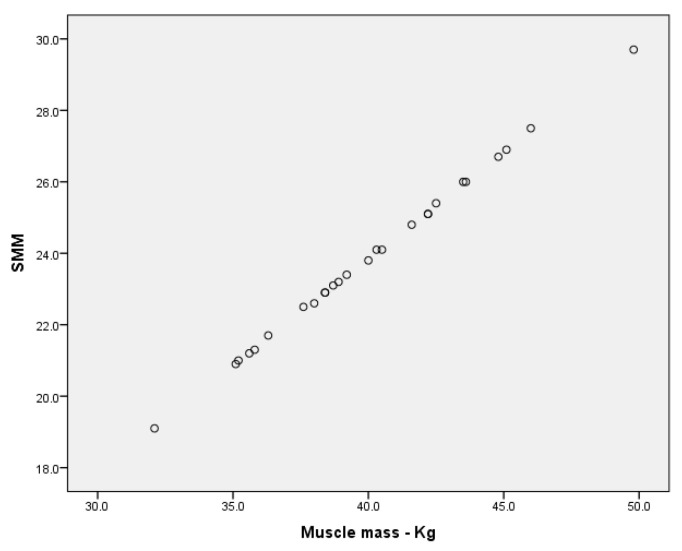
Correlation between SMM and muscle mass (kg) for the group of girls without ID.

**Figure 11 children-09-00582-f011:**
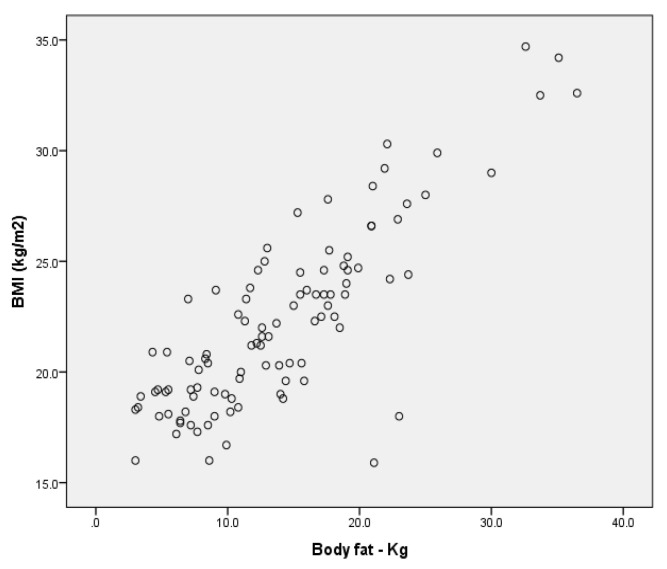
Correlation between BMI and body fat per group.

**Figure 12 children-09-00582-f012:**
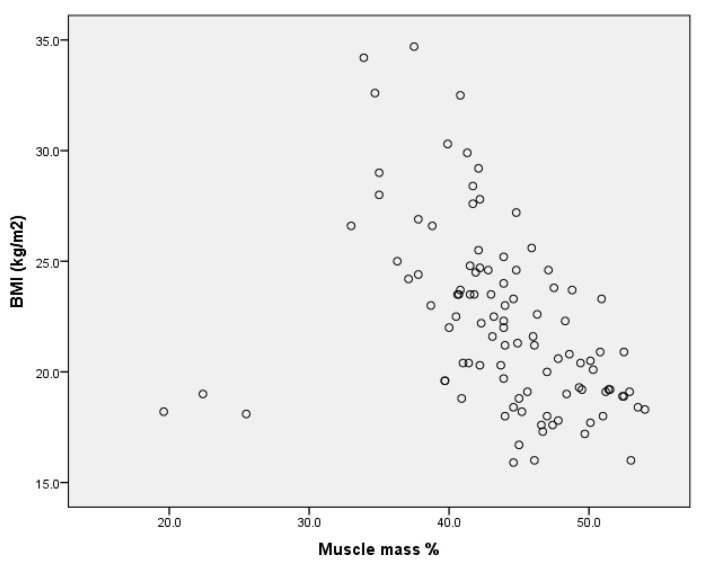
Correlation between BMI and percentage muscle mass per group.

**Table 1 children-09-00582-t001:** Repartition of the subjects by age, casuistry, and educational institutions.

Subjects	Gender	N	Age (mean ± std.dev.)	Casuistic Observation	Educational Institution
Group 1 (without ID) Without intellectual disability	M	23	17.7 ± 0.9	without ID	Mainstream education
Group 2 (without ID) Without intellectual disability	F	26	17.2 ± 0.9	without ID	Mainstream education
Group 3 (MID) Moderate intellectual disability	M	34	16.4 ± 0.9	MID	“Sf. Andrei” School Centre Gura Humorului; “Constantin Păunescu” School Centre Iaşi
Group 4 (MID) Moderate intellectual disability	F	12	16.2 ± 0.1	MID	“Sf. Andrei” School Centre Gura Humorului; “Constantin Păunescu” School Centre Iaşi
Group 5 (SID) Severe intellectual disability	M	6	16.8 ± 0.9	SID	“Sf. Andrei” School Centre Gura Humorului; “Constantin Păunescu” School Centre Iaşi

**Table 2 children-09-00582-t002:** The values of the distribution normality test (Shapiro–Wilk).

Group/Variable	Group 1	Group 2	Group 3	Group 4	Group 5
Age—years	0.795	0.275	0.476	0.322	0.035
Height—cm	0.558	0.506	0.873	0.000 **	0.109
Body mass—Kg	0.054	0.012 *	0.348	0.636	0.847
BMI (kg/m^2^)	0.122	0.015	0.005 *	0.310	0.538
Body fat %	0.288	0.099	0.200	0.170	0.411
Muscle mass %	0.378	0.096	0.000 **	0.007 *	0.324
BMR (kcal)	0.055	0.481	0.318	0.747	0.707
Body fat—Kg	0.048	0.006 *	0.003 *	0.724	0.733
Muscle mass—Kg	0.093	0.96	0.514	0.276	0.847
SMM	0.201	0.967	0.501	0.250	0.945

* Significance threshold *p* < 0.05; ** Significance threshold *p* < 0.001.

**Table 3 children-09-00582-t003:** Synthetic table with morphological parameters.

Variable		N	Mean	Std. Deviation	Std. Error	Minimum	Maximum
BMI (kg/h^2^)	G 1	23	22.957	4.1716	0.8698	17.7	32.5
	G 2	26	21.581	3.7966	0.7446	15.9	34.2
	G 3	34	22.538	4.3805	0.7512	17.2	34.7
	G 4	12	21.325	3.2897	0.9497	17.3	26.6
	G 5	6	22.333	5.3166	2.1705	16.0	29.0
	Total	101	22.231	4.0903	0.4070	15.9	34.7
Body fat %	G 1	23	16.148	7.1465	1.4902	4.6	27.8
	G 2	26	25.912	5.4527	1.0694	18.6	40.0
	G 3	34	19.144	8.2409	1.4133	6.6	38.7
	G 4	12	25.767	6.7294	1.9426	17.5	41.7
	G 5	6	23.333	12.6438	5.1618	6.0	38.0
	Total	101	21.240	8.3611	0.8320	4.6	41.7
Muscle mass %	G 1	23	47.022	4.6759	0.9750	36.3	54.0
	G 2	26	41.942	3.0833	0.6047	33.9	46.1
	G 3	34	44.326	7.2026	1.2352	19.6	52.9
	G 4	12	40.442	6.8489	1.9771	22.4	46.7
	G 5	6	43.500	7.2042	2.9411	35.0	53.0
	Total	101	43.816	6.0662	0.6036	19.6	54.0
BMR(kcal)	G 1	23	1887.09	274.216	57.178	1496	2726
	G 2	26	1400.54	128.759	25.252	1170	1726
	G 3	34	1744.26	226.088	38.774	1376	2165
	G 4	12	1393.92	60.237	17.389	1288	1489
	G 5	6	1549.00	152.548	62.277	1364	1757
	Total	101	1635.08	281.966	28.057	1170	2726
Body fat—Kg	G 1	23	12.548	8.0549	1.6796	3.0	33.7
	G 2	26	15.719	5.8224	1.1419	8.6	35.1
	G 3	34	13.074	7.9548	1.3642	3.4	36.5
	G 4	12	14.900	4.9171	1.4195	7.7	23.0
	G 5	6	15.500	10.1931	4.1613	3.0	30.0
	Total	101	13.996	7.2982	0.7262	3.0	36.5
Muscle mass—Kg	G 1	23	56.809	8.6844	1.8108	41.9	83.2
	G 2	26	40.054	4.0181	0.7880	32.1	49.8
	G 3	34	49.300	7.3912	1.2676	32.9	62.7
	G 4	12	39.467	3.3611	0.9703	34.1	44.0
	G 5	6	44.833	4.4907	1.8333	38.0	50.0
	Total	101	47.196	9.1887	0.9143	32.1	83.2
SMM	G 1	23	34.752	6.1352	1.2793	25.0	49.5
	G 2	26	23.885	2.4054	0.4717	19.1	29.7
	G 3	34	29.400	4.3789	0.7510	19.6	37.3
	G 4	12	23.192	1.8362	0.5301	20.4	25.8
	G 5	6	26.667	2.5820	1.0541	23.0	30.0
	Total	101	28.299	5.9250	0.5896	19.1	49.5

**Table 4 children-09-00582-t004:** The Post Hoc analysis for the morphofunctional parameters for groups of boys.

Dependent Variable	(I) Group	(J) Group	Mean Difference (I–J)	Std. Error	Sig.	95% Confidence Interval	
						Lower Bound	Upper Bound
Height—cm	1	3	7.1893 *	2.3681	**0.010 ***	1.498	12.880
		5	8.6304	4.0208	0.089	−1.033	18.293
	3	5	1.4412	3.8839	0.927	−7.893	10.775
Body mass—Kg	1	3	7.4680	3.8683	0.139	−1.828	16.764
		5	9.5406	6.5682	0.321	−6.244	25.325
	3	5	2.0725	6.3446	0.943	−13.175	17.320
BMI (kg/m^2^)	1	3	0.4183	1.1858	0.934	−2.431	3.268
		5	0.6232	2.0134	0.949	−4.215	5.462
	3	5	0.2049	1.9448	0.994	−4.469	4.879
Body fat %	1	3	−2.9963	2.2491	0.383	−8.401	2.409
		5	−7.1855	3.8189	0.153	−16.363	1.992
	3	5	−4.1892	3.6889	0.496	−13.054	4.676
Muscle mass %	1	3	2.6953	1.7261	0.270	−1.453	6.843
		5	3.5217	2.9308	0.457	−3.522	10.565
	3	5	0.8265	2.8310	0.954	−5.977	7.630
BMR (kcal)	1	3	142.822	64.809	0.079	−12.93	298.57
		5	338.087 *	110.043	**0.009 ***	73.63	602.54
	3	5	195.265	106.296	0.166	−60.19	450.72
Body fat—Kg	1	3	−0.5257	2.2140	0.969	−5.847	4.795
		5	−2.9522	3.7593	0.713	−11.987	6.082
	3	5	−2.4265	3.6313	0.783	−11.153	6.300
Muscle mass—Kg	1	3	7.5087 *	2.0805	**0.002 ***	2.509	12.508
		5	11.9754 *	3.5325	**0.004 ***	3.486	20.465
	3	5	4.4667	3.4122	0.396	−3.734	12.667
SMM	1	3	5.3522 *	1.3473	**0.001 ***	2.114	8.590
		5	8.0855 *	2.2876	**0.002 ***	2.588	13.583
	3	5	2.7333	2.2097	0.436	−2.577	8.044

* Significance threshold *p* < 0.05.

## Data Availability

All relevant data are within the study, and raw data are available on request.

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
