# Peer review of "The Analysis of the Correlations between BMI and Body Composition among Children with and without Intellectual Disability"

_children, 2022, doi:10.3390/children9050582_

Round 1

Reviewer 1 Report

Thank you for allowing me to review your work, it is always a pleasure and an opportunity to learn. The topic that the authors deal with is very interesting and they are right when they say that this type of studies do not abound, so, in principle, the chosen topic seems relevant to me.

The summary aims to compare between the group of children with and without intellectual disabilities, but in results and conclusion (within the abstract) this is not referred to.

The introduction provides sufficient background  but it could include accurate references about intelectual disability and body composition.

Although data are collected from 5 different centers, I do not understand the separation into 5 groups, since I do not see that there is a difference between groups 1 and 2 on the one hand and 3 and 4 on the other. If the objective of the research is to establish the differences between children with and without intellectual disabilities, these comparison groups are not designed according to this characteristic, but to their center of belonging, so their results and conclusions will not be able to achieve the objectives of their study or test their hypotheses; if the authors want to evaluate the influence of the existence or not of intellectual disability, together with the degree of said disability and gender, all together, so they should state it in the design of the research (objectives and hypotheses) or modify the analyses carried out.

The authors perform a normality test in which they show that the samples do not conform to normal for all variables and groups, but subsequently, they use parametric statistics. In my opinion this is not appropriate without a good justification. They could either show by histograms and Levenne test that curves are normal according to the considerations of some authors that they should justify, or use non-parametric tests. In the specific case of Anova, there is evidence about its use in samples that do not conform to normality, they could also use it if they justify it documentarily, but, in any case, that justification would be lacking.

There is no discussion section, I think it is essential.

Format:

  • Remove the zeros to the left of the point
  • In tables some p with uppercase and some lowercase
  • In the table of post hoc analyses it would be useful to indicate statistically significant results, for example, in bold (although they are already marked with *, it is more visual)

Reviewer 2 Report

I thank the authors for having been able to revise their writing, on a very interesting topic. However there are several points to fix.

The introduction is too narrow, should be expanded. Lines 75-78 could be added in the introduction. And in lines 83-85 there is a repetition (with lines 75-78). Lines 79-82 are not necessary. 

Matherial and methods must be improve. 

Was not measured the weight? and TANITA had specific equations for the adolescents? Statistical analysis could be in a specific part. Lines 132-137 should be in the introduction part. How were the participants found? What were the criteria for inclusion or exclusion? How has disability been assessed? ID, MID what they mean? There is not an explanation. Participants have the same age? What is their mean age? 

The results part is not clear. The analysis of normality is not well presented. In my knowledge is not necessary to report table 1. In addition, if there are variables that are not normal distributed, they must be analyzed with other kind of statistic analysis. Because it is not possible to use ANOVA for variables that are nor normal distributed. In h2, the 2 should be as an appendix. In the results, comments or consideration are not necessary. Please remove all comments, considerations and references to literature. Table 4 is not clear, I suggest to change the rows with the columns. There are too many graphs and they are not so necessary. Please remove or add only in the supplementary material. 

The discussion part of the results is completely missing. It must be added, otherwise the article is incomplete. 

Round 2

Reviewer 1 Report

  •  The objective of the research is still to establish the differences between children with and without intellectual disabilities, but the comparison groups remain not designed according to this characteristic, so their results and conclusions will not be able to achieve the objectives of their study or test their hypotheses. If the authors want to evaluate the influence of the existence or not of intellectual disability, together with the degree of said disability and gender, all together, (as they state in their reply they semm it is important) they should change the design of the research (objectives and hypotheses) and modify the analyses carried out.
  • Regarding normality; the autors give a reply, but the maniscript ramains the same. 
  • Discussión:
    • Discussion consists on a compilation of results, a comment about Levenne and 5 lines disscusing the results. Discussion is an essecntial part of a manuscript, and it must be better performed.
    • They introduce some comments aoubt Levenne test in discussion; I think it is not the apropriate place.

Reviewer 2 Report

I thank the authors for the changes they have made, surely now the article is more complete than the first version.

However, the part of the discussions is still almost completely missing. The authors have added a part that unfortunately is not enough. The most important results of the article should be reported in the discussions and compared with the existing literature, but they are almost completely missing here.

Round 3

Reviewer 1 Report

The authors have made the modifications required in the review.

Reviewer 2 Report

I thank the authors for having followed the suggestions and for having extended the Discussion part